# Universal Human Pose Representation for Multi-Modal Active Sensing

## Abstract

We propose *UniversalPose*, a unified pose estimation framework that supports a wide range of sensing modalities, including WiFi, mmWave, acoustic, LiDAR, and depth. While recent methods have explored such alternative modalities to improve robustness in situations where conventional RGB-based approaches often fail (*e.g.*, in low-light or occluded environments) and have privacy issues, they typically rely on modality-specific architectures, which limit their scalability and generalization to new sensor types. UniversalPose addresses these limitations by transforming all inputs into a shared representation of token sequences, enabling a single architecture to handle heterogeneous data formats. To ensure efficient and stable learning, we introduce pseudo-3D positional embeddings and apply multi-modal locality-aware self-attention, even for modalities without explicit spatial coordinates. Moreover, adopting such a modality-agnostic representation allows multi-modal fusion via simple token concatenation, which improves performance without architectural modifications. Extensive experiments demonstrate that UniversalPose achieves comparable or superior accuracy to modality-specific expert models while supporting multiple modalities through joint training. Moreover, with synchronized multi-modal inputs, the same architecture outperforms the existing state-of-the-art fusion model. Our code will be publicly available.

## 1 Introduction

Human pose estimation is a long-standing focus in machine learning, enabling a wide range of real-world applications such as augmented/virtual reality (AR/VR) Jiang & Ithapu (2021), anomaly detection Markovitz et al. (2020), and human–robot interaction Ye et al. (2021). So far, the field has relied extensively on models trained with RGB video data Xu et al. (2022); Sun et al. (2019). However, RGB-based methods frequently degrade in occluded environments or under low-light conditions Lee et al. (2023). Moreover, RGB imagery can carry rich personal identifiers, such as facial or contextual cues, that raise privacy concerns in surveillance and other sensitive scenarios.

To address these limitations, recent studies have begun exploring additional sensing modalities beyond RGB, including millimeter-wave (mmWave) radar Yang et al. (2023); Fan et al. (2025), WiFi Ren et al. (2022); Jiang et al. (2020), depth imaging Yang et al. (2023); Liu et al. (2025), spatial audio Shibata et al. (2023); Oumi et al. (2024), and LiDAR Ren et al. (2024); Zheng et al. (2022). Each modality exhibits unique advantages and limitations: for instance, unlike RGB, mmWave and LiDAR maintain accuracy in darkness Yang et al. (2023); Ren et al. (2024), while WiFi exhibits strong occlusion tolerance by penetrating walls Ren et al. (2022); Jiang et al. (2020); moreover, acoustic sensing avoids capturing identifiable imagery and thus better preserves visual privacy Shibata et al. (2023); Oumi et al. (2024). As human pose estimation becomes increasingly relevant across diverse domains, the ability to seamlessly switch between modalities based on environmental or application demands is key to ensuring robust and adaptable performance Liu et al. (2025).

Aiming to utilize this complementary information among modalities, recent studies have also introduced multi-modal fusion methods Chen & Yang (2024); Li et al. (2022). However, fusion methods inherently require synchronized collection of multiple signals, which restricts the availability of training data and makes the inference setup costly. Moreover, the state-of-the-art fusion method Chen & Yang (2024) trains its modality-specific encoders on data collected from a single sensor type, which restricts both the size of available training sets and the diversity of learned representations. Furthermore,

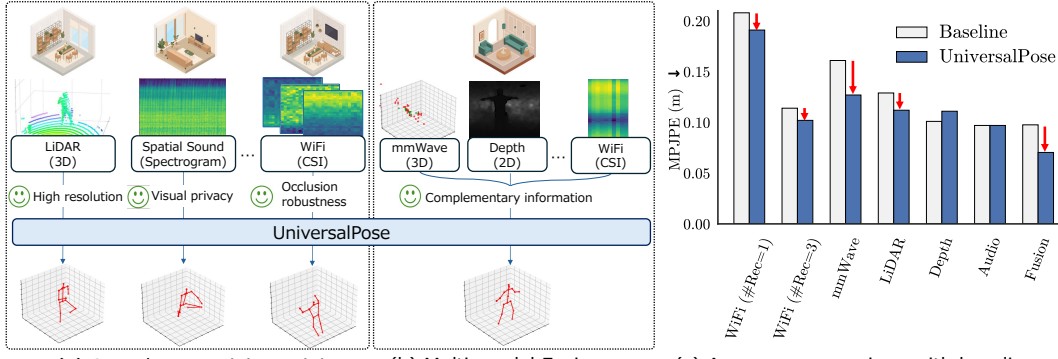

Figure 1: (a)(b) We propose **UniversalPose**, which employs a single encoder for multiple sensing modalities for human pose estimation. By sharing a common encoder, we perform joint training with diverse asynchronous datasets and multi-modal fusion with paired data. (c) Our model achieves higher accuracy than modality-experts/fusion baselines while handling diverse sensing modalities.

due to architectural discrepancies, a large-scale pre-training and fine-tuning paradigm—effective in computer vision Caron et al. (2021); He et al. (2022) and NLP Devlin et al. (2019); Achiam et al. (2023) domains—has not yet been realized for heterogeneous sensing in human pose estimation.

To overcome these challenges, we present **UniversalPose**, a new framework for human pose estimation that unifies multiple sensing modalities into a common representation space, enabling diverse signals to be processed within a single architecture (Fig 1 (a)). This is a challenging task, as it involves handling signals with fundamentally different formats and characteristics. For instance, 3D point clouds (LiDAR/mmwave) encode spatial geometry, whereas WiFi CSI and audio spectrogram capture frequency responses.

To handle these diverse signals, we convert every modality into a common tokenized representation and process all tokens with a single, modality-agnostic hierarchical Transformer encoder. In contrast to prior work Chen & Yang (2024) that relies on modality-specific encoders, we pair lightweight tokenizers with a shared transformer encoder, which improves scalability and yields a general-purpose model for heterogeneous signals. Specifically, for each modality, the tokenizer maps inputs to continuous tokens, which are then aggregated using self-attention with multi-modal local context. This encoder shares weights across modalities yet can be trained on each modality independently, so it does not require paired or synchronized recordings from the same environment. Moreover, its modality-agnostic design enables multi-modal fusion without architectural changes: tokens from different modalities can simply be concatenated and jointly processed.

To investigate the method's efficacy, we conduct large-scale experiments by integrating multiple datasets spanning **five sensing modalities and eight sensing settings**. By jointly training on **unpaired, asynchronous** data, UniversalPose demonstrates strong generalization across modalities, achieving comparable or superior performance to expert models tailored to individual signals. Furthermore, the same model architecture can be used to learn multi-modal fusion, achieving higher accuracy than the existing fusion method. Moreover, fine-tuning the pre-trained UniversalPose significantly improves accuracy when adapting to unseen environments or modalities.

We summarize our contributions as follows:

- We propose *UniversalPose*, the first Transformer–based backbone that can process point clouds, spectrograms, and other heterogeneous signals *within the same set of parameters*. This unification removes the need for modality-specific encoders and enables large-scale, cross-modal training.

- For stable joint multi-modal training, we introduce a locality-aware unified encoder that projects diverse modalities into a common spatial representation, using multi-modal local context.

- Our experiments demonstrate that *UniversalPose*, when jointly trained on **unpaired, asynchronous** data, (i) achieves **versatility across heterogeneous sensing modalities**, matching or surpassing expert per–modality models, and (ii) exhibits **cross-modal transferability**, enabling efficient fine-tuning to unseen modalities. In addition, when *synchronized* inputs are available, (iii) the same architecture delivers **synergistic multi-modal fusion**, leveraging complementary signal characteristics to attain higher accuracy than any single modality.

Table 1: Comparison of methods for handling diverse modalities.

| Method type | Available Information | Paired data | Cross-Modal Synergy | Scalability |
|---|---|---|---|---|
| **Single-modality** | Per-modality only (with trade-offs) | - | ✗ | Low |
| **Multi-modal Fusion** | Complementary | Required | ✓ | Low |
| **UniversalPose** | Complementary | Not Required (single) /Required (fusion) | ✓ | High |

## 2 RELATED WORK

Researchers have long pursued human pose estimation using a wide array of sensing signals. This section first introduces human pose estimation methods employing different modalities and highlight the key distinctions among them. We then discuss recent efforts in multi-modal fusion, which aim to leverage the complementary information provided by various sensor inputs, and identify the challenges that our work seeks to address as summarized in Table 1.

**Comparison of Modalities in Human Pose Estimation.** RGB video has been the most commonly used modality for human pose estimation Cao et al. (2019); Xu et al. (2022); Sun et al. (2019), fueled by advancements in deep convolutional neural networks and the availability of large-scale annotated datasets. These methods typically perform well in controlled lighting and unobstructed environments but struggle in scenarios with occlusions or poor visibility Lee et al. (2023). In response, alternative single-modality based approaches have emerged. LiDAR- Ren et al. (2024); Yang et al. (2023) or mmWave-based methods Xue et al. (2021); Zhao et al. (2018); Yang et al. (2023); Fan et al. (2025) typically emit laser pulses or radio waves, respectively, and measure reflections to construct a point cloud representation of the human body. While these approaches inherently capture depth information, they can suffer performance drops in adverse weather conditions such as rain or fog Chae et al. (2024). Depth image–based techniques, on the other hand, reduce the use of visually identifiable information compared to RGB, thereby offering enhanced privacy Yang et al. (2023); however, they have a narrower measurement range and remain susceptible to occlusions. WiFi-based methods exploit the ability of WiFi frequencies to penetrate walls Zhao et al. (2018); Ren et al. (2022), enabling potential coverage of larger areas. Yet, they are limited by lower spatial resolution and the need for multiple antennas. Finally, the most recent line of research utilizes acoustic signals for active sensing Yang et al. (2022); Shibata et al. (2023); Wang et al. (2024b); Oumi et al. (2024). Owing to their relatively long wavelength, sound-based methods offer privacy advantages over WiFi or mmWave, but their spatial resolution is significantly constrained.

**Multi-Modal Fusion.** Recently, multi-modal fusion approaches have gained increased attention, aiming to address the intrinsic challenges of each sensor modality and achieve more accurate human pose estimation. DeepFusion performs alignment and fusion of LiDAR and RGB image data, thereby enhancing robustness to noise and out-of-distribution samples in object detection Li et al. (2022). ImmFusion Chen et al. (2023) achieves accurate pose estimation in low-light and smoke/rain conditions by fusing RGB and mmWave data. A very recent work, X-Fi Chen & Yang (2024), introduces a flexible architecture that leverages cross-attention for various sensing signals, such as mmWave, WiFi, or RGB image data, enabling pose estimation with any combination of these modalities. This approach represents a notable advancement in multi-modal pose estimation. Although these methods achieve high-accuracy pose estimation, they require synchronized multi-modal datasets for training, which can be expensive and time-consuming to collect. Moreover, for example, X-Fi Chen & Yang (2024) freezes a pre-trained, modality-dependent encoder that was originally trained on a single modality. As a result, the encoder cannot exploit complementary information across modalities or benefit from large-scale data integration, potentially becoming a bottleneck in overall accuracy.

**Deep Learning Architectures for Heterogeneous Signals.** To broaden the applicability of models across diverse scenarios, recent research has explored unified architectures capable of handling tasks such as image/audio/3D object classification Girdhar et al. (2022); Srivastava & Sharma (2024), as well as tracking Wu et al. (2024), within a single model. Moreover, prior work Wang et al. (2024a) has shown that a shared model can enable scalable learning across robots with varying proprioceptive

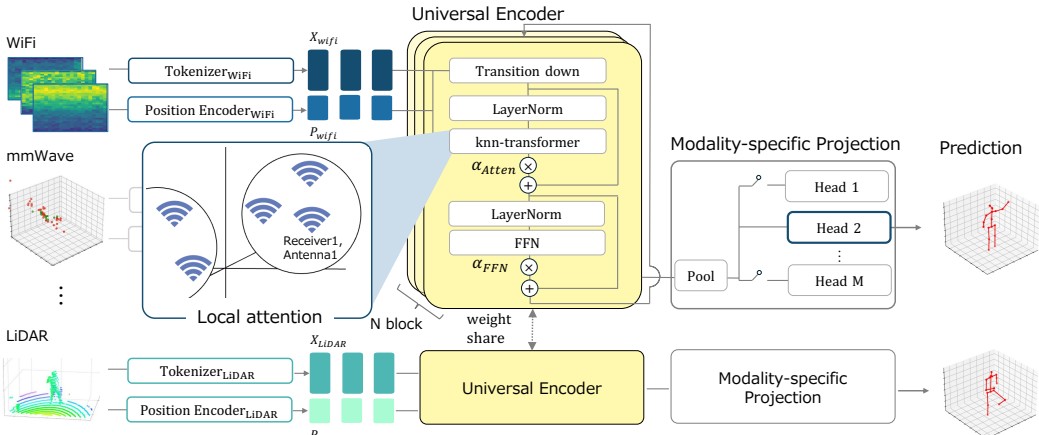

Figure 2: The proposed pipeline and details of the universal encoder. The proposed UniversalPose maps all input signals into a shared representation space using modality-specific tokenizers and positional encoders.

inputs and camera configurations, improving policy performance. However, such progress has not been observed in the field of active human sensing. A major obstacle lies in the heterogeneous characteristics and data formats of the sensing signals involved. Whereas RGB and depth images are generally two-dimensional, LiDAR and mmWave signals often appear as point clouds—and in some cases, mmWave can also be represented as a four-dimensional Range-Doppler-Azimuth-Elevation (RDAE) map. For these modalities, the data format typically corresponds to 3D space. By contrast, WiFi and spatial acoustics are frequently expressed in the frequency domain, for example, as channel state information (CSI) or spectrograms, respectively. Due to these format discrepancies, existing research has tended to employ separate neural network architectures—one encoder per modality Chen & Yang (2024); Dai et al. (2025). This practice not only restricts the size of the training dataset available for each individual encoder, but also necessitates retraining from scratch whenever a new signal is introduced or the sensing configuration (e.g., the number of antennas or microphones) changes, causing shifts in the input format. Such fragmentation significantly reduces efficiency and impedes scalability in real-world deployments.

## 3 METHODOLOGY

This section introduces **UniversalPose**, a framework for multi-modal human pose estimation. In §3.1, we present the overall design of UniversalPose, explaining how we consolidate diverse modalities into a single Transformer-based architecture to output 3D human pose. Next, §3.2 details our minimally designed tokenizers for each modality that transform raw sensor outputs into $d$-dimensional tokens. Finally, §3.3 describes how a shared universal encoder processes multi-modal tokens in a unified manner, stabilizing training by leveraging modality-specific context and learnable skip connections.

### 3.1 UNIVERSALPOSE: OVERVIEW

Fig 2 shows the overview of our pipeline. Our goal is to handle substantially different sensing signals, such as LiDAR, mmWave, WiFi, depth image, and acoustic, within a single unified model that outputs 3D human poses $\widetilde{\mathbf{p}}$, *i.e.*, UniversalPose : $\{\mathbf{X}_m, \mathbf{P}_m\} \mapsto \widetilde{\mathbf{p}}$. Here, $m$ denotes the modality index, $\mathbf{X}_m \in \mathbb{R}^{L_m \times d}$ and $\mathbf{P}_m \in \mathbb{R}^{L_m \times 3}$ are the continuous tokens and their positional embeddings, respectively. Here, $L_m$ denotes the number of tokens, and $d$ is the embedding dimension. The details of these inputs are discussed in the following section.

A key aspect of our design is to consolidate as much representational power as possible into a shared *Universal Encoder*, placing **over 90%** of the learnable parameters in the shared encoder, while minimizing the number of learnable parameters required for modality-specific processing, such as tokenizers or heads. This setup enables large-scale multi-modal training, leveraging any cross-modal synergies that arise. Through joint training and backpropagation, UniversalPose learns to produce a

single 3D human pose output $\widetilde{\mathbf{p}}$ that benefits from cross-modal information, even when trained on asynchronous, unpaired datasets.

## 3.2 Modality Specific Tokenizer

UniversalPose maps different modalities into a common sequence of tokens, allowing unified training across modalities. This section briefly describes the nature of each modality and explain how we embed its raw data into a $d$-dimensional token sequence with length $L_m$, yielding $\mathbf{X}_m \in \mathbb{R}^{L_m \times d}$. After tokenization, zero-padding is applied to unify the token sequence length across all modalities.

**LiDAR** Since Light Detection and Ranging (LiDAR) has dense 3D point clouds, directly converting each point into a token can be expensive. Therefore, we apply a small Point Transformer Zhao et al. (2021) module for downsampling before embedding the resulting points into $d$-dimensional tokens.

**mmWave** Millimeter-wave radar outputs sparser point clouds (often $< 50$ points), each storing 3D position, velocity, and reflection intensity. A MLP converts these $(x, y, z, v, I)$ features into tokens.

**WiFi** A single CSI (Channel State Information) frame forms a tensor of shape #time $\times$ #transmitters $\times$ #receivers $\times$ #antennas $\times$ #subcarriers. We embed the amplitude information from each subcarrier, mapping it into $d$-dimensional tokens via another small MLP.

**Acoustic** We use mel-spectrogram slices, where each slice corresponds to a time step and contains $C$ mel-frequency bins across $K$ directional microphone channels (e.g., $C = 128$). The $C$-dimensional vector at each time, direction pair is independently transformed by an MLP.

**Depth Images** Depth data are patchified, flattened, and projected into $d$-dimensional tokens.

## 3.3 Multi-modal Token Processing via Universal Encoder

In this section, we describe how the proposed Universal Encoder tackles the primary obstacle of our task, *the diversity of multi-modal data*, by stabilizing joint training and uniformly extracting pose information across all sensing modalities.

**Multi-modal hierarchical attention mechanisms.** In preliminary experiments, we observed that unconstrained global self-attention leads to significant representational divergence across modalities, resulting in severe performance degradation for certain modalities. To address this, our method harnesses modality-specific inductive biases (geometric, temporal, and antenna-/channel-wise) in a unified manner across sensing types via a multi-modal hierarchical attention scheme. This unified scheme also introduces implicit regularization of attention patterns, stabilizes joint training, and enables more efficient learning of shared representations.

To build a unified pipeline with high scalability, we learn hierarchical representations for each modality while minimizing per-modality specific engineering. For point-cloud modalities (e.g., mmWave, LiDAR) we exploit the spatial relationships among reflection points, whereas for modalities without explicit spatial geometry (e.g., WiFi, acoustic) we induce hierarchical structure from signal metadata (e.g., antenna/microphone IDs, directionality) in an end-to-end data-driven manner.

Specifically, we extract reflection-point coordinates or apply a learnable linear transformation to relevant metadata (e.g., antenna IDs, directional channels) to generate unified pseudo-3D coordinates:

$$\mathbf{p}_{m,i} = \begin{cases} (x_{m,i},\ y_{m,i},\ z_{m,i}) & \text{if } m \text{ is a point-cloud}, \\ \text{LinearProjection}_m(\text{metadata}_{m,i}) & \text{otherwise.} \end{cases} \tag{1}$$

This learnable linear transformation is trained using the loss on the pose outputs. At each transformer block, local self-attention is applied over each token's $k$ nearest neighbors in the resulting coordinate space, capturing spatial continuity for point clouds and temporal or directional structure for frequency-based modalities (see Fig. 2). For capturing long-range token dependencies, we also perform token downsampling by leveraging the transition-down module from Point Transformer Zhao et al. (2021).

**Further stabilization under data heterogeneity.** It is well known that training a deep neural network with a single parameter set for diverse tasks (e.g., via multi-task learning) can induce optimization instabilities and, in some case, degrade accuracy due to gradient conflicts or domain shift Wu et al. (2024); Xu et al. (2022). In our experiments, we observed that even with a relatively shallow number

of blocks, training could collapse for certain modalities. To further improve training stability, we adopt ReZero-based residual connections Bachlechner et al. (2021), which make the skip pathways learnable (see $\alpha_{\text{FFN}}$ and $\alpha_{\text{Atten}}$ in Fig 2). This design allows the model to behave like a shallow network at the early stages of training, enabling more stable joint learning across modalities. To further stabilize training, we introduce Layer Normalization to mitigate the heterogeneity in input distributions.

**Other model details.** After applying $N$ locality-aware self-attention blocks, we perform simple pooling across the token dimension. The resulting features are then passed through a modality-dependent MLP heads to produce the final pose prediction $\widetilde{\mathbf{p}}$. We optimize this prediction by minimizing the mean squared error (MSE) loss between $\widetilde{\mathbf{p}}$ and the ground-truth pose $\mathbf{p_{gt}}$. In contrast to the original Point Transformer Zhao et al. (2021), we additionally incorporate a feed-forward network (FFN) to enhance the model's capacity for nonlinear representation.

**Parameter-sharing oriented design.** While UniversalPose includes lightweight *modality-specific* components (the tokenizers and the heads), the *vast majority* of parameters lie in the shared Universal Encoder. A quantitative parameter breakdown and comparison with the existing work Chen & Yang (2024) are shown in Appendix A.1. Making most weights modality-independent enables joint training across merged datasets and efficient fine-tuning for new modalities using the pretrained weights.

## 4 EXPERIMENTAL SETTINGS

**Dataset.** To evaluate the performance of our model across multiple sensing modalities, we employed four distinct human pose estimation datasets. First, we used MM-Fi Yang et al. (2023), which is currently the largest multi-modal pose estimation dataset. MM-Fi captures human poses under four different environmental settings, making use of multiple sensing signals. From these signals, we selected four fundamentally different modalities—depth, LiDAR, mmWave, and WiFi. Second, we included Person-in-WiFi 3D Yan et al. (2024), where human poses are measured with **multiple** WiFi receivers—unlike MM-Fi, which utilizes a **single** receiver setup. To align with the settings of other datasets, we used samples featuring a single subject only. Third, we incorporated the dataset from Shibata et al. (2023), herein referred to as TSP2Pose, which uses an acoustic chirp for active pose estimation. Finally, we adopted BGM2Pose Shibata et al. (2025), a dataset where pose estimation is conducted under dynamically changing background music. Note that BGM2Pose is recorded in a different room from TSP2Pose. Moreover, the acoustic sensing signals in these two datasets differ substantially: TSP2Pose employs a noisy chirp source, whereas BGM2Pose uses ordinary stereo background music. We label the MM-Fi environments as E01–E04, the TSP2Pose as E05, the Person-in-WiFi 3D as E05, and the BGM2Pose as E07, respectively. Table 7 in Appendix A.4 summarizes the environments, modalities, dataset sources, and the number of frames used.

**Baseline models.** We compare our model against the following modality-specific baselines: (i) mmWave: Point Transformer Zhao et al. (2021) (M), (ii) LiDAR: Point Transformer Zhao et al. (2021) (L), (iii) WiFi: MetaFi++ Zhou et al. (2023) (Single/Multi), (iv) Audio (chirp): TSP2Pose Shibata et al. (2023), and (v) Audio (BGM): BGM2Pose Shibata et al. (2025). Note that we specify Point Transformer (M/L) for mmWave and LiDAR because the two signals differ substantially in point density and feature composition (e.g., velocity and intensity). Consequently, the downsampling ratio and input dimensions of the fully connected layers must be adjusted. Similarly, we adopt two separate MetaFi++ (Single/Multi) because the number of receivers varies across datasets. Some models partially emulate the idea behind UniversalPose and serve as flexible multi-modal unified baselines. Point Transformer (M&L) uses a modality-dependent MLP with zero-padding to handle both mmWave and LiDAR, while MetaFi++ (Single&Multi) adopts a similar strategy to accommodate varying antenna counts in WiFi data. We also compare our model with these two. For multi-modal fusion, we compare against the current state-of-the-art X-Fi Chen & Yang (2024). Although X-Fi is also transformer-based, it relies on frozen modality-specific expert encoders.

**Evaluation Metrics.** We evaluate 3D human pose estimation accuracy using two standard metrics: Mean Per Joint Position Error (MPJPE) and Procrustes-Aligned MPJPE (PA-MPJPE). MPJPE computes the average Euclidean distance between predicted and ground-truth joint positions. PA-MPJPE further removes rigid transformations via Procrustes alignment, isolating structural pose accuracy. Due to space constraints, PA-MPJPE results are shown in Table 9 in Appendix A.6. All numerical values are reported in meters.

Table 2: Ablation study.

| Model | Training | WiFi (Sing) | mmWave | LiDAR | Depth | TSP | WiFi (Mul) | Avg |
|-------|----------|-------------|--------|-------|-------|-----|------------|-----|
| ViT-like | Joint | 0.205 | 0.225 | 0.170 | 0.297 | 0.145 | 0.204 | 0.208 |
| Ours | Separate | 0.222 | 0.194 | 0.186 | 0.212 | 0.132 | 0.103 | 0.175 |
| Ours | Joint | 0.215 | 0.179 | 0.188 | 0.227 | 0.138 | 0.123 | 0.178 |
| Ours w/o ReZero | Joint | 0.208 | 0.185 | 0.175 | 0.207 | 0.138 | 0.161 | 0.179 |

Table 3: Accuracy comparison against single-modality baselines.

| Model | WiFi (Sing) | mmWave | LiDAR | Depth | TSP | WiFi (Mul) |
|-------|-------------|--------|-------|-------|-----|------------|
| MetaFi++ (Single) | 0.205 | - | - | - | - | - |
| Point Transformer (M) | - | 0.161 | - | - | - | - |
| Point Transformer (L) | - | - | 0.129 | - | - | - |
| A2J | - | - | - | **0.101** | - | - |
| TSP2Pose | - | - | - | - | **0.097** | - |
| MetaFi++ (Multi) | - | - | - | - | - | 0.114 |
| MetaFi++ (Single & Multi) | 0.212 | - | - | - | - | 0.208 |
| Point Transformer (M&L) | - | 0.214 | 0.115 | - | - | - |
| Ours | **0.191** | **0.127** | **0.112** | 0.111 | **0.097** | **0.102** |

**Implementation Details.** The hyperparameters for baseline models were determined based on the prior works Yang et al. (2023); Zhou et al. (2023); Shibata et al. (2023; 2025). The number of blocks in UniversalPose was set to $N = 6$. Our proposed method was trained for only 30 epochs using SGD. Our model was trained using a single NVIDIA A6000 GPU. Please refer to the Appendix A.5 for more details, including learning rate, optimizer settings, and batch size.

## 5 Experiments and Results

**Evaluation Settings.** We evaluate the performance of our UniversalPose and existing baselines under three testing settings: **(1) Asynchronous Joint Pre-training:** training uses *unpaired, asynchronous* data from multiple environments, and testing is on environments/modalities seen during joint training; **(2) Cross-Signal Generalization:** evaluates generalization to *unseen* environments and *novel* modalities by fine-tuning the jointly trained *UniversalPose* on the new input ; **(3) Multi-Modal Fusion:** When synchronized multi-modal recordings are available for both training and evaluation, we train the same architecture for fusion and measure accuracy gains from cross-modal synergy.

For evaluation, we selected two training subjects from the datasets E01–E05. Since subject annotations are not available for E06 Yan et al. (2024), we instead use their train/val split from room 'S11'. For ablative and exploratory experiments, we conducted small-scale training using all available data from E06 and two randomly selected training subjects from other datasets (E01–E05). In contrast, for the main results, we used all training data except for the validation subjects (see Appendix A.5).

**Ablation and Exploratory Study.** To validate the effectiveness of our training strategy for active 3D human pose estimation across multiple sensing modalities, we conduct an ablation study summarized in Table 2, where we systematically vary the model architecture and training procedure. These results are based on the experimental setting (1). The middle two rows of the table show that our proposed method, under a single joint-training run, delivers consistently high accuracy across many modalities and, for several of them, further improves performance through joint training.

As an architectural baseline, we implement a VisionTransformer-like model Dosovitskiy et al. (2020) that first converts all modalities into continuous token sequences and then feeds them into an encoder composed of multiple self-attention and FFN blocks. For point cloud–based modalities, explicit positional encoding cannot be directly defined. Therefore, we adopt the same projection used in our proposed method to compute positional embeddings from modality-specific metadata or 3D positions. Note that this model has a comparable number of parameters to UniversalPose (approximately 20M). The table results indicate that while this model is capable of learning strong representations for certain modalities (e.g, wifi(i), LiDAR), it also leads to training collapse or degraded performance in others (e.g., mmwave, WiFi(ii)). In contrast, our proposed model consistently achieves high accuracy across a wide range of modalities, demonstrating its strong adaptability to diverse input signals. We

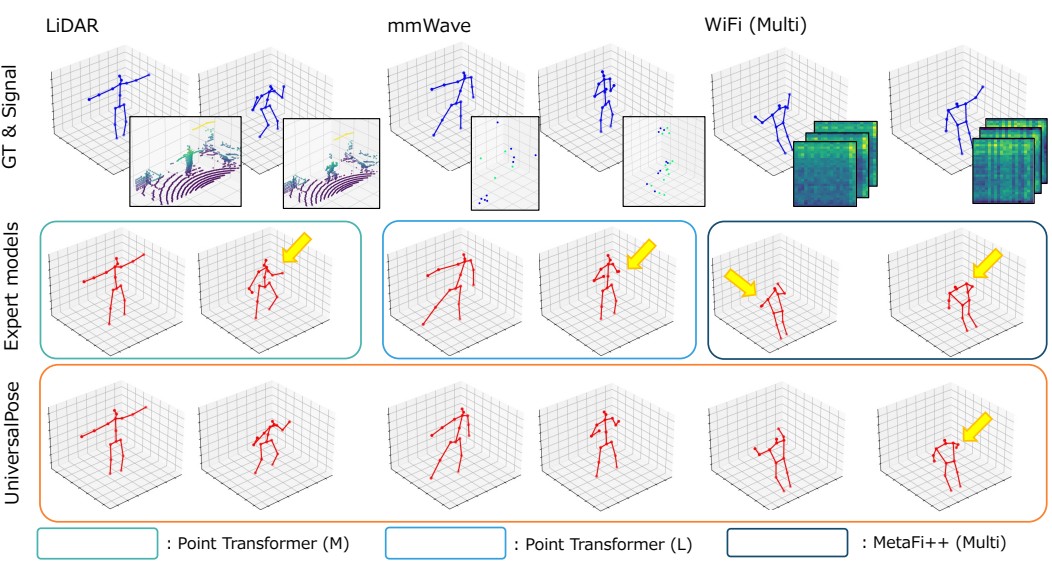

Figure 3: Qualitative results. Additional results are included in Appendix A.6.

Table 4: Effect of multi-modal joint-pretraining.

| Model | Depth (E04) | | Music (E07) | |
|---|---|---|---|---|
| | PE | PA-PE | PE | PA-PE |
| A2J | 0.101 | 0.059 | — | — |
| BGM2Pose | — | — | 0.111 | 0.059 |
| Ours (scratch) | 0.126 | 0.080 | 0.145 | 0.080 |
| Ours (pretrained) | 0.115 | 0.077 | 0.136 | 0.075 |

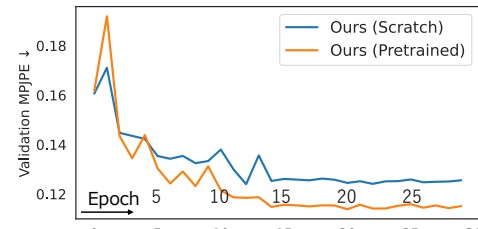

Figure 4: Depth sensing training.

speculate that the stability of our method arises because restricting attention to local neighborhoods acts as implicit regularization and prevents large shifts in latent representations across modalities.

**Accuracy comparison in the in-training setting.** We scale up training by using all subject data from E01 to E06, and compare our model against the baselines. Table 3 compares the baseline models and our proposed *UniversalPose* under the (i) in-training setting. In this scenario, the UniversalPose framework is jointly trained using Single/Multi-receiver WiFi (E01/06), mmWave (E02), LiDAR (E03), Depth (E04), and Acoustic (TSP) (E05), while each single-modal expert model is trained solely on data from its respective environment.

These results demonstrate that our model achieves the highest accuracy across most sensor types, even though it is not specifically tailored to any single signal. Note that existing expert models are specialized for a single signal type, rendering them incompatible with other modalities and thus leaving many entries blank in the table. These findings suggest that a more general model with signal tokenization and a unified Transformer-based architecture, combined with expanded training data, delivers higher generalization performance than specialized signal-specific processing.

Fig. 3 illustrates qualitative comparisons for three different sensing signals. We observe that expert baselines, constrained by limited training data, produce substantially erroneous pose estimations (see yellow arrows). Moreover, particularly in the LiDAR case, the baseline model sometimes yields anatomically implausible predictions (e.g., a distorted neck). We conjecture that, due to the limited diversity of subjects in the baseline model's training set, it fails to generalize adequately and collapses when encountering unknown subjects. For additional qualitative results using other sensing signals, please refer to Appendix A.6.

**The Effect of Unified Pre-training.** To evaluate the generalization capability of our model to unseen modalities and the effectiveness of fine-tuning, Table 4 compares the performance of modality-specific baselines and UniversalPose, both with and without fine-tuning on unseen modality data. Specifically, we designated depth (E04) and music (E07) as unseen modalities and excluded these modalities and corresponding environments from the training set of UniversalPose. The "pretrained" variant in

Table 5: **Multi-modal fusion results.** Left: comparison with X-Fi (we follow their official train/val split and action protocol). Right: accuracy improvement via fusion on Environment 01 (cross-subject).

| Multi-modal fusion comparison | | | |
|---|---|---|---|
| Model | Modalities | MPJPE | PA-MPJPE |
| X-Fi | D+L+R+W | 0.098 | 0.047 |
| Ours | D+L+R+W | **0.064** | **0.044** |

*D = Depth, L = LiDAR, R = RF (mmWave), W = WiFi.*

| Accuracy improvement via fusion (Env. 01) | | | |
|---|---|---|---|
| Model | Modalities | MPJPE | PA-MPJPE |
| Ours | D | 0.114 | 0.067 |
| Ours | L | 0.087 | 0.056 |
| Ours | R | 0.132 | 0.072 |
| Ours | W | 0.190 | 0.096 |
| Ours | D+L+R+W | **0.072** | **0.050** |

Table 4 refers to the model pre-trained on all modalities except the target one (i.e., E04 for depth and E07 for music), and subsequently fine-tuned using only target modality data.

As baselines, we used A2J for depth and BGM2Pose Shibata et al. (2025) for music, each trained solely on its respective environment (i.e., E04 and E07). As shown in Table 4, although UniversalPose was not specifically designed for the music modality, it achieves accuracy comparable to the modality-specialized BGM2Pose. Moreover, for both depth and music, the fine-tuned variant of UniversalPose consistently outperforms its from-scratch counterpart, highlighting the effectiveness of multi-modal pretraining. This improvement is further supported by the learning curves in Figure 4, which clearly demonstrate lower final error with pre-trained UniversalPose. These results suggest that large-scale multi-modal pretraining can serve as a powerful alternative to modality-specific engineering, offering a scalable and generalizable framework for diverse sensing conditions.

**Multi-Modal Fusion.** We trained and evaluated UniversalPose in a multi-modal fusion setting and compare it with the state-of-the-art method X-Fi Chen & Yang (2024). **Without any architectural modifications**, we concatenate the outputs of each modality-specific tokenizer (together with positional encodings) along the token dimension and feed the resulting sequence to the Universal Encoder. For a fair comparison with X-Fi, we follow their official training/validation split and action protocol. The results are shown in Table 5.

Unlike X-Fi, which includes RGB during training and applies random modality dropout, we train UniversalPose only on the four active-sensing modalities considered in this work (D, L, R, W) and do not use modality dropout (all four modalities are provided at each training instance). Despite this difference, our model substantially outperforms X-Fi on both metrics, indicating that the proposed fusion mechanism is highly effective within the unified architecture.

To further quantify the benefit of fusion inside our framework, we conduct an ablation on E01 under a cross-subject protocol, comparing single-modality against fused inference. As summarized in Table 5, fusing all four modalities yields a marked improvement over any individual modality, highlighting effective cross-modal synergy captured by the local attention in the pseudo-coordinate space.

**Other results.** We do not use explicit modality alignment (such as contrastive learning) when training UniversalPose. Still, optimizing for human pose estimation yields shared representations; Fig. 5 in Appendix A.2 shows these representations mixing across modalities as the number of seen samples grows. As for the pseudo-3D representation derived from meta-information (Sec. 3.3), we present visualization results and interpretation on acoustic signals using the trained model in Appendix A.3.

# 6 CONCLUSION

In this work, we presented UniversalPose, a novel Transformer-based framework that consolidates multiple sensing modalities, i.e., LiDAR, mmWave, WiFi, depth, and spatial acoustics within a single architecture for human pose estimation. Our design employs minimal, modality-specific tokenizers to convert each sensor's raw data into a unified token format, followed by a shared hierarchical encoder that merges these tokens into a common representation. Our experiments show that UniversalPose (i) enables scalable joint training on asynchronous, cross-environment datasets to deliver versatile inference across modalities, (ii) transfers efficiently to unseen modalities with fine-tuning, and (iii) when synchronized signals are available, achieves substantial accuracy gains through simple multi-modal fusion within the same architecture. As future work, we will pursue parameter-efficient adaptation for new modalities and establish a single training regime that unifies asynchronous joint training with multi-modal fusion, aiming for a more general-purpose model.

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

# A  APPENDIX

## A.1  EXTENDED ANALYSIS: ARCHITECTURE COMPARISON WITH A MODALITY-DEPENDENT DESIGN

This appendix contrasts UniversalPose with a representative *modality-dependent* approach (X-Fi). Both pipelines include modality-specific processing, but a key distinction lies in the **proportion of modality-specific parameters** relative to the total. As shown in Table 6, X-Fi allocates a large portion of trainable parameters to modality-specific encoders and heads, whereas UniversalPose **shares over 90% of parameters across modalities**, keeping only lightweight tokenizers, positional encodings, and output heads modality-specific.

Table 6: **Number of modality-specific parameters and parameters shared across modalities (in millions).** Percentages indicate the proportion of parameters that are modality-specific within the processing pipeline. A lower percentage indicates that fewer parameters are modality-specific within the processing pipeline, which is desirable for a versatile model.

| Method | LiDAR | mmWave | WiFi-CSI | Depth | Shared |
|--------|-------|--------|----------|-------|--------|
| X-Fi | 17.07 (89%) | 4.60 (68.7%) | 24.43 (92.1%) | 14.45 (87.3%) | 2.10 |
| Ours | 1.53 (8.0%) | 0.18 (1.0%) | 0.20 (1.0%) | 0.27 (1.4%) | 17.06 |

This difference yields several practical advantages:

- **Fine-tuning scalability.** Because modality-dependent encoder architectures cannot support cross-modal fine-tuning, adapting to a new modality typically requires redesigning and training most of the encoder from scratch. In contrast, UniversalPose reduces cross-modal adaptation to training *from scratch* only the lightweight components (tokenizers, positional encoders, and heads), while the shared backbone is *fine-tuned* rather than reinitialized.
- **Simpler training pipeline.** Modality-dependent designs usually require separately pretraining each modality-specific encoder in advance. UniversalPose jointly trains all components, including the shared universal encoder, within a unified framework.
- **Improved data efficiency.** Joint training improves performance for underrepresented modalities (e.g., mmWave and depth) by letting them benefit from shared representations learned from richer modalities; sharing the encoder effectively increases the usable training data and improves generalization.
- **Better scalability.** As new modalities are added, modality-dependent pipelines grow parameters roughly linearly. In contrast, UniversalPose keeps modality-specific weights minimal and maintains most capacity in the shared backbone, which is better suited for multi-modal large-scale training.

## A.2  LATENT FEATURE VISUALIZATION

Fig. 5 visualizes the output of the final Transformer block of our proposed *UniversalPose*, projected into two dimensions using t-SNE. The visualization is based on inference results for held-out validation subjects. Panel (a) corresponds to the model trained on a restricted subset of training subjects—specifically, all available data from E06 and two randomly selected subjects from each annotated dataset (E01–E05)—as in the ablation study (Table 2). Panel (b) shows the model trained with all available training subjects across all datasets. Comparing (a) and (b), we observe that using all training subjects, thereby increasing the number of seen samples, leads to a latent space where features from different modalities are more tightly interwoven (especially, see red/orange/green dots). This comparison suggests that, even without employing explicit strategies such as contrastive learning for multi-modal alignment, the proposed *UniversalPose* implicitly aligns modalities over the course of training by solving the shared task of human pose estimation with a single universal encoder.

## A.3  PSEUDO-3D COORDINATES VISUALIZATION

Fig. 6 visualizes the learned pseudo-3D coordinates for Spatial Audio (TSP) metadata (channel and time) produced by the trained *UniversalPose* model. Here, different colors denote channels (acoustic

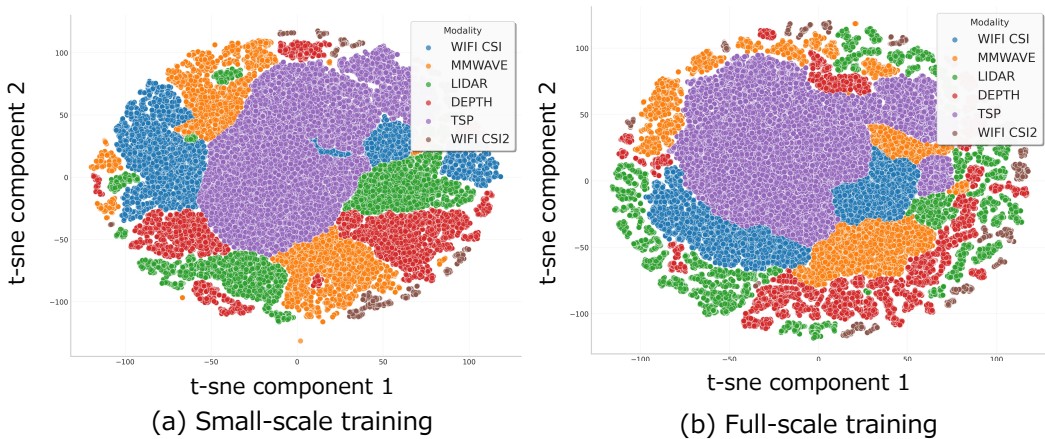

Figure 5: Latent Feature visualization.

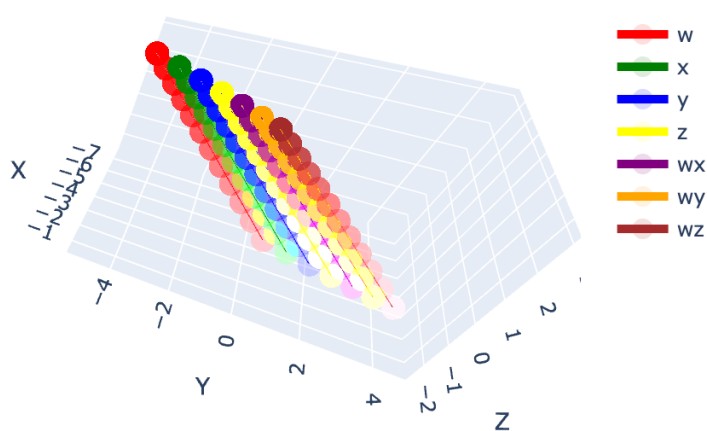

Figure 6: Pseudo 3D coordinates for spatial audio (TSP) signals.

B-Format + intensity vector), while color intensity encodes time within the sequence (t=0. . . 12). We map per-token metadata $m_i \in \mathbb{R}^M$ (e.g., for audio: channel and timestamps; for CSI: timestamp, receiver ID and antenna ID) to a 3D coordinate $p_i \in \mathbb{R}^3$ via a *learnable linear* transformation $p_i = Wm_i + b$, trained end-to-end through the pose-estimation loss. Because this mapping is linear and order-preserving, the continuity of the original metadata is retained while each axis can be rescaled through learning, causing temporally (or spectrally) adjacent tokens to remain close in the pseudo-3D space and enabling efficient capture of local dynamics. Crucially, these coordinates define the neighborhoods for $k$NN attention so the attention graph adapts to each modality's inductive structure. For example, if the temporal axis is scaled up, each token's neighbors will more likely be tokens with different directionality IDs but similar timestamps. In this way, simple metadata guides a shared $k$NN-attention backbone to exploit modality-specific structure without architectural changes or synchronized multi-modal data during training.

## A.4 MULTI-MODAL HUMAN SENSING DATASET

This work uses the datasets summarized in Table 7. Note that for the E06 *person-in-wifi 3D* dataset we use only room "S11" and restrict evaluation to single-person sequences.

Table 7: Detailed information about hyperparameters. LR means learning rate and BS means batch size.

| Env | Modality | Frame Count |
|-----|----------|-------------|
| E01 | WiFi (single receiver) | 30888 |
| E02 | mmWave | 30247 |
| E03 | LiDAR | 30583 |
| E04 | Depth | 30709 |
| E05 | Audio (chirp) | 55922 |
| E06 | WiFi (multi receiver) | 13101 |
| E07 | Audio (music) | 51320 |

## A.5 THE EXPERIMENTAL CONFIGURATION

**The UniversalPose configuration.** In our *UniversalPose* model, self-attention was performed with a dimensionality of 512. The modality-specific tokenizers are composed of a 2-layer MLP. The modality-specific heads are implemented as 3-layer MLPs. The transformation module that maps meta-information into a pseudo-3D coordinate space is implemented as a single linear layer. The Transformer block used for computing local attention consists of 6 blocks, each with a hidden dimension of 512. For LiDAR inputs, we applied a two-layer Point Transformer module as a preprocessing step. Additionally, in the kNN-based self-attention mechanism employed in *UniversalPose*, the value of $k$ was set to 32. Each block in UniversalPose employs a feed-forward network (FFN) consisting of a linear transformation, a ReLU activation, LayerNorm for improved stability, and a final fully connected linear layer. In the multi-modal joint training scheme adopted by UniversalPose, mini-batches were constructed using vanilla sampling, with the sampling ratio proportional to the amount of data in each dataset.

**Model Hyperparameters.** Table 8 summarizes the hyperparameters used to train both the baseline models and our proposed model. We primarily adopted the hyperparameter values from the original works. However, in cases where training was unstable under our setting, particularly when only a single environment was available for each single-modality experiment, we performed additional tuning. Here, WiFi-CSI (i) refers to the single-antenna setting from the MMFi dataset Yang et al. (2023), while WiFi-CSI (ii) corresponds to the multi-antenna setting in the Person-in-WiFi 3D dataset Yan et al. (2024).

Table 8: Detailed information about hyperparameters. LR means learning rate and BS means batch size.

| Model | Modality | Epochs | LR | Optimizer | Scheduler | BS |
|-------|----------|--------|-----|-----------|-----------|-----|
| Point Transformer | mmWave | 50 | 0.001 | AdamW | None | 64 |
| Point Transformer | LiDAR | 50 | 0.01 | SGD | MultiStepLR V2 | 24 |
| TSP2Pose | TSP (audio) | 100 | 0.01 | SGD | MultiStepLR V2 | 256 |
| MetaFi++ | WiFi-CSI (i) | 100 | 0.05 | SGD | MultiStepLR V1 | 36 |
| MetaFi++ | WiFi-CSI (ii) | 100 | 0.05 | SGD | MultiStepLR V2 | 32 |
| A2J | Depth | 50 | 0.00035 | Adam | MultiStepLR V2 | 64 |
| BGM2Pose | BGM (audio) | 30 | 0.05 | SGD | MultiStepLR V3 | 48 |
| Ours | ALL | 30 | 0.05 | SGD | MultiStepLR V3 | 48 |

We use `torch.optim.lr_scheduler.MultiStepLR` with different milestone and decay configurations for each scheduler variant:

**MultiStepLR V1** The learning rate is decayed by a factor of 0.5 at epochs 20, 40, 60, and 80. This setting is used for long training runs (e.g., 100 epochs).

**MultiStepLR V2** The learning rate is reduced by a factor of 0.1 at epochs 20 and 40. This configuration is applied in mid-length training (e.g., 50–100 epochs).

**MultiStepLR V3** The learning rate is decreased by a factor of 0.1 at epochs 15 and 25. This variant is used for shorter training schedules (e.g., 30 epochs), including the UniversalPose training setup.

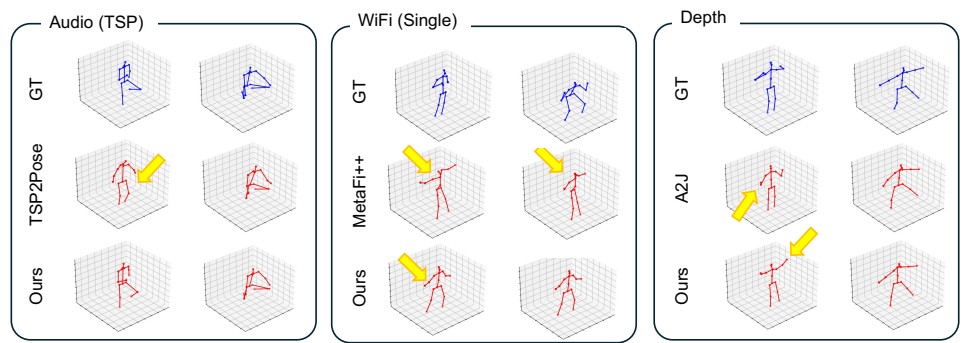

Figure 7: Qualitative Results on TSP (Chirp audio), WiFi (ii), and Depth Image

Table 9: Accuracy comparison based on PA-MPJPE.

| Model | WiFi (Single) | mmWave | LiDAR | Depth | TSP | WiFi (Multi) |
|---|---|---|---|---|---|---|
| MetaFi++ (Single) | 0.120 | - | - | - | - | - |
| Point Transformer (M) | - | 0.069 | - | - | - | - |
| Point Transformer (L) | - | - | 0.074 | - | - | - |
| A2J | - | - | - | **0.059** | - | - |
| TSP2Pose | - | - | - | - | 0.092 | - |
| MetaFi++ (Multi) | - | - | - | - | - | 0.087 |
| MetaFi++ (Single&Multi) | 0.212 | - | - | - | - | 0.118 |
| Point Transformer (M&L) | - | 0.081 | 0.050 | - | - | - |
| Ours | **0.093** | **0.062** | **0.045** | 0.076 | **0.065** | **0.081** |

Furthermore, although TSP2Pose Shibata et al. (2023) and BGM2Pose Shibata et al. (2025) leverage auxiliary losses—such as adversarial loss and contrastive loss—to further enhance accuracy, we excluded these components during training to enable a fair comparison based on simple architectural differences. Additionally, while the original implementations of these models adopt a sequence-to-sequence prediction scheme, we modified their output heads to perform frame-by-frame predictions to align with the other modalities.

The number of predicted joints was standardized to 17, following the MMFi dataset Yang et al. (2023). For the Chirp Audio Shibata et al. (2023) and BGM datasets Shibata et al. (2025), any additional joints were removed prior to training and evaluation to ensure consistency.

**Data Settings for Training and Evaluation.** This section provides detailed information on which portions of each dataset were used for training and evaluation. The MMFi dataset Yang et al. (2023) provides two types of motion data: daily activities and rehabilitation exercises. Following their protocol 2, we used only the rehabilitation motions for training and evaluation in this work. In both the MMFi Yang et al. (2023) and TSP2Pose Shibata et al. (2023) datasets, two subjects per room were used for evaluation. For the Person-in-WiFi 3D dataset Yan et al. (2024)—which lacks explicit subject ID information-we used the train/eval split provided by the official implementation and performed evaluation under the single-person setting (room 'S11'). For the BGM sensing setting Shibata et al. (2025), we used the music track "ARNOR", with one subject reserved for evaluation and the remaining four subjects used for training.

### A.6 Qualitative Results on Additional Modalities

Fig. 7 presents qualitative results on three additional modalities, chirp audio, multi-antenna WiFi, and depth, which could not be included in the main paper. Across many modalities and poses, the proposed method qualitatively produces outputs that are closer to the ground-truth poses (see yellow arrows indicating errors).

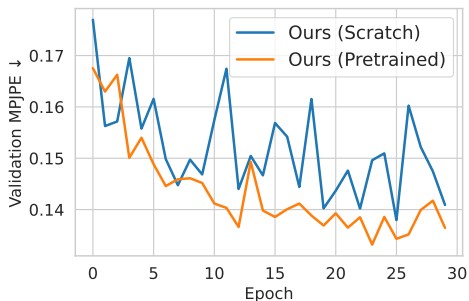

Figure 8: The effect of joint pre-training for BGM sensing

## A.7 QUANTITATIVE RESULTS BASED ON PA-MPJPE

Due to space limitations, the accuracy based on PA-MPJPE (Procrustes Aligned Mean Per Joint Position Error) could not be included in the main text. Table 9 summarizes these results. These results demonstrate that the proposed UniversalPose either outperforms or matches single-modality expert models across a wide range of sensing modalities.

## A.8 EFFECT OF PRETRAINING IN THE BGM SENSING SETTING

As shown in Table 5 and Figure 5 of the main paper, *UniversalPose* enables multi-modal joint pre-training and fine-tuning by handling all modalities with the same architecture and parameter set. This unified design allows the model to improve performance even on unseen modalities. Figure 8 presents the training curve for the BGM dataset, which was not included in the main paper. Although this is a challenging setting involving dynamic sensing signals, which introduces some instability during training, the performance improvement achieved through pretraining is clearly observed.

## A.9 DISCLOSURE ON THE USE OF LARGE LANGUAGE MODELS (LLMS)

For the preparation of this manuscript, we used a large language model (LLM) in a supportive manner, specifically to assist with English grammar correction.

