# OpenReview forum: "Universal Human Pose Representation for Multi-Modal Active Sensing"
_ICLR.cc/2026/Conference — Submitted to ICLR 2026_

### Official Review · Reviewer_G2H2 · 2025-10-16

**Soundness:** 3
**Presentation:** 3
**Contribution:** 3
**Rating:** 6
**Confidence:** 2

**Summary:**

The paper introduces UniversalPose, which is a unified framework for 3D human pose estimation that supports a wide range of non-RGB sensing modalities, including WiFi, mmWave, acoustic, LiDAR, and depth. The core idea is to overcome the limitations of modality-specific architectures, which suffers from poor scalability and difficulty with large-scale pre-training. Experiments across seven sensing settings demonstrate that UniversalPose achieves accuracy comparable to or better than specialized, single-modmodality expert models.

**Strengths:**

- Clearly written paper and easy to follow.
- Propose a strong unified model that achieve SOTA performance across multiple benchmarks.
- The model consistently achieves comparable or superior human pose estimation performance to highly specialized, single-modality baselines, validating the effectiveness of the general-purpose, unified representation.
- The results on cross-signal generalization via fine-tuning also strongly support the pre-training paradigm.

**Weaknesses:**

- Given the extreme differences between modalities (e.g., mmWave vs. WiFi tensor), the used tokenizers bear a critical burden or loading in transforming representation. A deeper analysis for the minimal design of these tokenizers, especially considering their role in mapping complex raw data into the unified token space can enhance the insights from this work.
- Tab.2 shows that joint training slightly degrades performance for some modalities compared to separate training, a discussion or analysis of representational trade-offs during the joint training process would strengthen the paper.
- The major differences with X-Fi is shifting the parameters to an universal encoder. Why deos architecture like X-Fi only support synchronized multimodal datasets for training?

**Questions:**

I hope the author can address my concerns in weakness.

---

> ### Author Response · Authors · 2025-11-23
>
> We thank the reviewer for the positive assessment of our work and for recognizing the effectiveness of UniversalPose as a unified model that achieves strong performance across multiple benchmarks and benefits from cross-signal pre-training. Below we address the weaknesses (W) raised by the reviewer.
>
> **(W.1) On the design of the tokenizers.**
> We thank the reviewer for this valuable suggestion. In the current implementation, each tokenizer is intentionally kept minimal and consists of a 2-layer MLP, which we view as a near-minimal design for mapping raw signals into the unified token space. In the revised version, we will provide an ablation in the appendix that compares this 2-layer MLP with (i) a single fully connected layer and (ii) a deeper MLP, in order to more clearly quantify the impact of tokenizer capacity on performance.
>
> **(W.2) On representational trade-offs under joint training.**
> We appreciate the reviewer’s suggestion to further analyze representational trade-offs. As shown in Appendix Fig.5(a)(b), even after joint training with the UniversalEncoder, features from different modalities still form distinct clusters, indicating that modality-specific structure is preserved. In the revised version, we will extend this analysis by more carefully visualizing the feature distributions across modality pairs and discussing which combinations in joint training lead to performance gains, in order to better characterize these trade-offs.
>
> **(W.3) On why X-Fi only supports synchronized multimodal training.**
> X-Fi focuses on how to fuse encoded features from each modality. It uses *frozen* modality-specific encoders that are pretrained, and learns only the fusion module on top of these fixed features. As a result, it requires synchronized multimodal datasets, because there are no trainable parameters that can be updated from unpaired single-modality data. In other words, X-Fi is designed for the setting where synchronized multimodal recordings are available, and extending it to fully asynchronous single-modality data would require redesigning parts of the training pipeline, which is outside the scope of that work.

---

> > ### Comment · Reviewer_G2H2 · 2025-11-24
> >
> > The author's response has addressed my concerns, and I will keep my positive ratings. Good luck!

---

### Official Review · Reviewer_cPmu · 2025-10-27

**Soundness:** 1
**Presentation:** 2
**Contribution:** 2
**Rating:** 2
**Confidence:** 4

**Summary:**

The paper presents UniversalPose, a Transformer-based framework for human pose estimation across multiple sensing modalities (WiFi, mmWave, LiDAR, acoustic, depth). Using a shared encoder for asynchronous, unpaired multimodal training is an interesting approach, and the evaluation includes comprehensive single-modal baselines. While some modality-specific components remain and certain design choices (e.g., ReZero residuals, asynchronous training) lack full empirical justification, the work demonstrates reasonable cross-modality generalization.

**Strengths:**

• The paper proposes UniversalPose, a unified Transformer-based framework capable of processing heterogeneous sensing modalities (WiFi, mmWave, acoustic, LiDAR, depth) under a single shared encoder, leveraging asynchronous, unpaired datasets for training.

• A sufficient number of single-modal baselines (e.g., MetaFi++, Point Transformer, TSP2Pose, A2J) are included to validate that UniversalPose maintains or surpasses modality-specific performance while using shared parameters.

• The work demonstrates strong cross-modality generalization through joint pre-training and fine-tuning, achieving competitive or better accuracy even on unseen modalities (e.g., BGM audio) compared to modality-specialized networks.

**Weaknesses:**

•	The paper claims to “unify all modalities within a single set of parameters and remove the need for modality-specific encoders,” yet the model still depends on modality-specific tokenizers, positional encoders and projection heads. Thus, it does not entirely remove modality-specific components, and the true scalability gain is probably overstated.

•	Although the model shows improved fusion results on the fixed D+L+R+W combination, it lacks experiments on varying modality combinations (e.g., D+R, L+W) or tests of generalization to unseen modality pairs (e.g. train on D+L+R+W and inference on R+W).

•	There is no direct comparison with fusion frameworks tailored for specific modality combinations (e.g., RGB–LiDAR fusion in autonomous driving). These limit understanding of whether the proposed universal encoder is competitive in specialized tasks.

**Questions:**

•	How does a simple MLP-based tokenizer outperform or match modality-specific encoders in extracting discriminative modality-dependent features?

•	What is the motivation for embedding the ability to exploit complementary information across modalities directly within the shared encoder, rather than employing dedicated fusion or cross-modal interaction blocks as in prior multimodal frameworks? From an intuitive perspective, what advantages does this integrated design offer in terms of representation learning or scalability?

•	Considering the substantial discrepancies among modalities, gradient conflicts during asynchronous joint training are a common challenge in multimodal learning. In this context, what are the advantages of employing a shared universal encoder to extract pose features across heterogeneous modalities? Moreover, while the authors attribute training stability to the introduction of ReZero-based residual connections, the ablation results in Table 2 line 329 and 330 show only marginal improvements, which do not convincingly substantiate the effectiveness of this design. To strengthen the argument for ReZero’s contribution, the authors could visualize training loss curves with and without ReZero to illustrate convergence stability.

•	In the proposed model, each modality is equipped with its own projection head. During multi-modal fusion, how are these modality-specific heads handled when multiple modalities are concatenated? Is a unified prediction head employed, or is there a joint optimization scheme that integrates outputs from different heads?

•	The three training settings ((1) asynchronous joint pre-training, (2) cross-signal generalization, (3) fusion) are insufficiently detailed. How are batches sampled across asynchronous datasets? How is the modality imbalance handled during optimization?

•	When combining various modality embeddings of different scales and distributions via simple token concatenation, how does the model prevent imbalance or dominance by specific modalities?

•	For experiments in table 4, what amount of data was used to fine-tune the pretrained model on the unseen domain, and what specific training procedure does the “from-scratch” variant refer to? Furthermore, what is the underlying intuition or rationale for expecting that pretraining on other modalities can effectively complement the adaptation process to an unseen modality, given the substantial information gap that exists between different sensing modalities?

•	The paper emphasizes the advantage of unpaired, asynchronous training, yet experiments only show same-modality or same-combination tests. How does asynchronous single-modality training concretely enhance multi-modal inference without synchronization?

**Minor suggestions:**

•	In Table 4, it would be better to explicitly define PE and PA-PE as MPJPE and PA- MPJPE, respectively.

•	In Table 7, the caption is misaligned with the table content.

---

> ### Author Response · Authors · 2025-11-24
>
> We thank the reviewer for finding our unified approach “interesting approach” and for recognizing that our proposed method can handle diverse sensing modalities. Below we address the weaknesses (W) and the questions (Q) raised by the reviewer.
>
> **(W.1) On Modality-Specific Components and Parameter Sharing.**
> We agree with the reviewer that UniversalPose still includes modality-specific tokenizers and prediction heads. However, we emphasize that these components are deliberately designed to be minimal, and the core contribution of UniversalPose lies in the extensive parameter sharing across modalities. As described in Section 3.1 and Table 6, **more than 90%** of the total parameters are shared across all modalities through a single unified encoder.
>
> Importantly, Table 4 demonstrates that by leveraging these shared parameters and performing fine-tuning together with a small number of modality-specific parameters trained from scratch, UniversalPose can improve performance across different sensing modalities. This indicates that knowledge transferred through the shared weights effectively contributes to improving performance across modalities. Therefore, although a new modality may require training a lightweight tokenizer and head, the majority of the model remains **reusable**, which significantly differs from conventional pipelines that require training a separate model per modality.
>
> **(W.2) On Fusion with Varying Modality Combinations.**
> We agree that it is important to evaluate fusion under varying modality combinations. We first note that, to the best of our knowledge, existing multi-modal fusion methods for human pose estimation (including X-Fi [1]) also do not explicitly demonstrate generalization to unseen modality pairs. In X-Fi (SOTA method for multi-modal fusion sensing), robustness to different modality subsets is achieved by applying modality dropout during training. We adopt the same strategy for UniversalPose: we train on the full set {D, L, R, W} with random modality dropout and then evaluate on the requested combinations L+W and D+R, as well as the full combination, using the MM-Fi dataset. The results are summarized in Table 1, where UniversalPose achieves comparable performance to X-Fi and, depending on the metric and modality pair, can even outperform it (e.g., lower MPJPE and PA-MPJPE for L+W). Combined with the fact that UniversalPose also supports asynchronous joint training and cross-modal fine-tuning, these results highlight the flexibility of our architecture across diverse modality configurations.
>
> _Table 1: Fusion performance comparison between UniversalPose and X-Fi._
>
> | Modalities        | MPJPE↓ (Ours) | MPJPE↓ (X-Fi) | PA-MPJPE↓ (Ours) | PA-MPJPE↓ (X-Fi) |
> |-------------------|:-------------:|:-------------:|:----------------:|:----------------:|
> | L + W (Requested) | **0.091**     | 0.160         | **0.059**        | 0.103            |
> | D + R (Requested) | 0.101         | **0.098**     | 0.069            | **0.047**        |
> | D + R + W + L     | **0.090**     | 0.098         | 0.061            | **0.047**        |
>
>
>
> [1] X.Chen et al, "X-FI: A MODALITY-INVARIANT FOUNDATION MODEL FOR MULTIMODAL HUMAN SENSING", in ICLR2025
>
> **(W.3) On Comparison with Specialized Fusion Frameworks.**
> We agree that direct comparison with fusion frameworks tailored to specific modality combinations would be informative. However, to the best of our knowledge, in the human pose estimation setting considered in this work there are no customized methods that jointly exploit multiple modalities such as LiDAR, mmWave, WiFi, depth, and acoustic signals. As noted in the abstract and Section 1 of the main paper, our target applications assume non-RGB modalities due to privacy considerations. While there exist models specialized for combinations like RGB+LiDAR or RGB+depth, we are not aware of any methods that use multiple modalities drawn solely from the set we focus on. If the reviewer is aware of an appropriate baseline, we would greatly appreciate the pointer and will make every effort to include a comparison with that method within the revision period.

---

> ### Author Response · Authors · 2025-11-24
> **Answers for questions (Part1)**
>
> We would like to thank the reviewer for the many detailed questions and for giving us the opportunity to clarify these points. Below, we provide responses to each question (Q1–Q8).
>
> **(Q1) How does a simple MLP-based tokenizer outperform or match modality-specific encoders in extracting discriminative modality-dependent features?**
> Our intention is not to claim that a small MLP alone is more expressive than carefully engineered modality-specific encoders. The tokenizer’s role is to map heterogeneous raw signals into a common token space, while the shared UniversalEncoder is responsible for most of the discriminative feature extraction. Please refer to (Q2), (Q3), and (Q7) for a discussion of the advantages of using a shared encoder, rather than separate encoders, to extract pose-discriminative features.
>
> **(Q2) Why embed complementary cross-modal interaction directly in the shared encoder instead of using dedicated fusion blocks, as in prior multimodal frameworks?**
> Our architecture based on a shared UniversalEncoder offers three main advantages.
>
> First, joint training increases the amount of data available for learning the encoder. If each modality had its own encoder and fusion were handled by a separate module, the effective training data for each encoder would be limited to a single modality. In contrast, UniversalPose trains a single encoder on all modalities, allowing it to benefit from the union of all datasets. As shown in Table 2, several modalities achieve better accuracy under joint training than when trained separately.
>
> Second, using one shared encoder reduces both model size and training cost. Since we do not need to maintain and train a separate encoder per modality, the overall parameter count is smaller and we avoid running multiple independent training procedures.
>
> Third, sharing the encoder across modalities enables cross-modal fine-tuning. As demonstrated in Table 4, we can adapt UniversalPose to an unseen modality by fine-tuning the shared encoder together with a small number of modality-specific parameters, instead of training a dedicated encoder from scratch for every new sensing modality. This cross-modal fine-tuning pipeline fundamentally relies on having a single encoder architecture that is shared across modalities; if each modality used a different encoder design, the parameters could not be directly reused or adapted in this way, and a separate model would need to be trained for each new modality.
>
> **(Q3) On the shared universal encoder design and the role of ReZero.**
> We thank the reviewer for raising this point. Regarding the advantages of a shared universal encoder under asynchronous joint training, Appendix Figure 5(a)(b) shows that, even after joint training, features from different modalities still form distinct clusters, indicating that modality-specific structure is well-preserved. At the same time, comparing Figure 5(a) and 5(b) reveals that training on more scaled data causes these clusters to partially mix, suggesting that UniversalPose learns both (i) modality-specific pose representations and (ii) modality-invariant, human pose-centric representations. This is consistent with Table 2, where joint training improves accuracy for several modalities, and with Table 4, where the same Universal Encoder can be effectively fine-tuned on unseen modalities, demonstrating **the practical utility of modality-agnostic human pose representations**.
>
> Concerning ReZero and training stability, its effect appears less in the average numbers and more in preventing severe failures on specific modalities. In Table 2, the “w/o ReZero” variant leads to a large performance drop for WiFi (Multi), whereas the full model with ReZero maintains stable performance across all modalities. Since the goal of UniversalPose is to serve as a generalist model that handles many modalities simultaneously, we adopt ReZero to avoid such modality-specific collapses and to keep joint training robust.
>
> **(Q4) In the proposed model, each modality is equipped with its own projection head. During multi-modal fusion, how are these modality-specific heads handled when multiple modalities are concatenated?**
> For the multi-modal fusion setting, we use a specific single prediction head. Concatenated tokens from all modalities are fed into the shared universal encoder, and only this fusion head is used for prediction; we do not combine outputs from the single-modality heads.

---

> ### Author Response · Authors · 2025-11-24
> **Answers for question (Part2)**
>
> **(Q5) The three training settings ((1) asynchronous joint pre-training, (2) cross-signal generalization, (3) fusion) are insufficiently detailed. How are batches sampled across asynchronous datasets? How is the modality imbalance handled during optimization?**
>
> (1) In asynchronous joint pre-training, each mini-batch is constructed by first sampling a dataset (modality) and then sampling random frames from it, where the sampling probability of each dataset is proportional to its size (no up/down-sampling).
>
> (2) In the cross-signal generalization setting, fine-tuning is performed only on the new modality (e.g., depth or BGM audio), so each mini-batch consists solely of samples from that modality.
>
> (3) For the fusion setting, we use the MM-Fi dataset, where all four modalities are paired. Each fusion batch contains samples with all modalities present, so there is no modality imbalance during optimization.
>
> **(Q6) When combining various modality embeddings of different scales and distributions via simple token concatenation, how does the model prevent imbalance or dominance by specific modalities?**
> In our current implementation, we do not apply any explicit re-weighting or rescaling across modalities at the token level. Although the number of tokens differs by modality (e.g., around 70 tokens for mmWave versus around 30 tokens for single-receiver WiFi), all tokens are simply concatenated and processed by the same Transformer blocks (UniversalEncoder).
>
> **(Q7) For experiments in Table 4, what amount of data was used to fine-tune the pretrained model on the unseen domain, and what does the “from-scratch” variant refer to? Why should pretraining on other modalities help an unseen modality despite the large gaps between sensing modalities?**
> In Table 4, both the “from-scratch” and “fine-tuning” settings use the same training split of the downstream dataset for the unseen modality (e.g., depth or BGM audio). Please refer to Table 7 in the appendix for the amount of data used in each downstream task. The only difference lies in how the model is initialized. In the “from-scratch” setting, a UniversalPose model with the same architecture is trained solely on this downstream dataset from **random initialization**. In the “fine-tuning” setting, we instead initialize from the jointly **pretrained UniversalPose** and then fine-tune the shared encoder together with a small number of modality-specific parameters on the same downstream data.
>
> Although the sensing modalities differ, the underlying task is always 3D human pose estimation. Pretraining on other modalities allows the encoder to learn generic priors about human pose (e.g., kinematic structure and typical motion patterns), as well as how pose changes manifest in different signal domains—for example, how pose dynamics appear in the frequency domain (joint training on TSP, downstream on BGM) or how they affect depth structure (joint training on LiDAR, downstream on depth). Fine-tuning can then adapt these shared priors to the new modality, which we believe explains the consistent performance gains over training from scratch.
>
> **(Q8) How does asynchronous single-modality training concretely enhance multi-modal inference without synchronization?**
> In our current experiments, unpaired asynchronous joint pre-training and fusion training are conducted as separate experiments. Our goal is to demonstrate that the same UniversalPose architecture can be directly applied in both settings, handling asynchronous single-modality datasets as well as synchronized multi-modal inputs (e.g., MM-Fi) without changing the backbone design.
>
> It is also possible to combine unpaired asynchronous training and fusion training within a single training run. We are running such experiments and will report the corresponding results in the appendix of the revised version.
>
> Once again, thank you very much for all the detailed questions. I apologize for the lack of explanation, and we will make sure to include this information in the camera-ready version or the supplementary material.

---

> > ### Author Response · Authors · 2025-11-27
> > **Inference Results on More Diverse Modality Combinations.**
> >
> > Motivated by the reviewer’s valuable feedback, we extended our multi-modal fusion experiments with UniversalPose to evaluate performance under **a wider range of modality combinations**. As noted above, by applying modality dropout during training, a single training run enables the model to handle a wide range of modality combinations at inference time. The results we previously reported were obtained after only 5 training epochs due to time constraints; here, we retrain both UniversalPose and X-Fi for 30 epochs, consistent with the main paper setting. Table 2 compares UniversalPose and X-Fi across many modality combinations.
> >
> > **Table 2: Modality-fusion Comparison of UniversalPose and X-Fi (Unit: m)**
> >
> > | Modalities | MPJPE (Ours) | MPJPE (X-Fi) | PA-MPJPE (Ours) | PA-MPJPE (X-Fi) |
> > | :--- | :---: | :---: | :---: | :---: |
> > | D | **0.0841** | 0.1018 | 0.0578 | **0.0484** |
> > | L | **0.0702** | 0.1671 | **0.0466** | 0.1032 |
> > | R | **0.121** | 0.1274 | **0.0611** | 0.0698 |
> > | W | **0.196** | 0.2256 | **0.105** | 0.1053 |
> > | D+L | **0.0704** | 0.1025 | **0.0484** | 0.0484 |
> > | D+R | **0.0766** | 0.0980 | 0.0523 | **0.0473** |
> > | L+R | **0.0692** | 0.1098 | **0.0459** | 0.0634 |
> > | D+W | **0.0844** | 0.1018 | 0.0581 | **0.0481** |
> > | L+W | **0.0699** | 0.1595 | **0.0466** | 0.1027 |
> > | R+W | **0.113** | 0.1172 | **0.0620** | 0.0627 |
> > | D+L+R | **0.0695** | 0.0960 | 0.0476 | **0.0473** |
> > | D+L+W | **0.0701** | 0.1020 | 0.0482 | **0.0481** |
> > | D+R+W | **0.0767** | 0.0970 | 0.0523 | **0.0471** |
> > | L+R+W | **0.0693** | 0.1074 | **0.0459** | 0.0631 |
> > | D+L+R+W | **0.0693** | 0.0976 | 0.0475 | **0.0474** |
> >
> > We observe that UniversalPose outperforms X-Fi on the majority of combinations and metrics. These results indicate that (i) the local self-attention mechanism with pseudo 3D coordinate in UniversalPose is effective for multi-modal fusion, and (ii) updating a shared encoder during fusion training, rather than optimizing only a fusion module on top of frozen encoders as in X-Fi, can lead to higher overall accuracy.
> >
> > Taken together, the following capabilities establish UniversalPose as a highly general and practical framework for active-sensing–based human pose estimation:
> >
> > * **Handling multiple modalities** via asynchronous joint training (Table 3 in main paper);
> > * **Improving accuracy on unseen modalities** through cross-modal fine-tuning (Table 4 in main paper); and
> > * **Performing effective multi-modal fusion** across diverse modality combinations (Table 2 above).

---

### Official Review · Reviewer_gjws · 2025-10-29

**Soundness:** 3
**Presentation:** 2
**Contribution:** 3
**Rating:** 4
**Confidence:** 3

**Summary:**

This paper introduces UniversalPose, a new Transformer–based human pose estimation framework, which unifies multiple sensing modalities (WiFi, millimeter wave, LiDAR, depth, and acoustics) into a shared token representation space, allowing for the use of a single architecture to handle different modalities.

**Strengths:**

1.The paper proposes a Transformer–based backbone network that can process multi-modal signals within the same parameter set and perform joint training on cross modal unpaired data.
2.The experiments are comprehensive, with evaluations including multiple datasets and baseline models, demonstrating Asynchronous Joint Pre-training, Cross-Signal Generalization, and Multi-Modal Fusion.

**Weaknesses:**

1.The author claims to use a single architecture to process multi-modal signals without the need for specialized encoders and architectures for each modality, but in fact, each modality still needs to be processed separately, such as using Point Transformer to process LiDAR, using different MLPs to process mmWave, WiFi, Acoustic, etc., and the obtained features need to be passed through modality dependent MLP heads to produce the final pose prediction. If a completely new modality is added, the tokenizer needs to be redesigned.
2.Although the paper focus on better utilizing multi-modal data, when UniversalPose only uses single-modal data, its performance are likely to degrade to below those of specialized expert models.
3. I don't think there's any advantage to using so many modalities. For the current task, more modalities isn't necessarily better. If using just one or two modalities can achieve the same effect as using all modalities, then I don't think the additional cost of using all modalities is justified.

Minor mistakes:
1.L303 ‘the Person-in-WiFi 3D as E05’ should be E06.
2.Figure 3: Point Transformer (M) and Point Transformer (L) are reversed.

**Questions:**

1.In Accuracy comparison in the in-training setting, although UniversalPose performs better than other expert models, training with multi-modal data results in a larger total amount of training data compared to training with single-modal data. A fairer baseline would be to train a UniversalPose instance separately for each modality (i.e. "Ours (Separate)"), but these results are only presented in the ablation experiment (Table 2) and there is insufficient direct comparison with the expert model. To what extent does the performance of this method stem from the unified "generalist" architecture itself, and to what extent does it stem from its ability to train with larger and more diverse amounts of data?
2.Are the ‘scratch’ in Table 4 and Figure 4 trained using only the target modality? Compared to ‘retrained’, ‘scratch’ uses less data for training? The scratch results here are worse than A2J and BGM2Pose. Does this further indicate that when UniversalPose only uses single-modal data, its performance will degrade to inferior to the expert model.

---

> ### Author Response · Authors · 2025-11-23
>
> We thank the reviewer for acknowledging the comprehensiveness of our experiments and for recognizing that our proposed method can handle diverse sensing modalities. Below we address the weaknesses (W) and the questions (Q) raised by the reviewer.
>
> **(W.1) On Modality-Specific Components and Parameter Sharing.**
> We agree with the reviewer that UniversalPose still includes modality-specific tokenizers and prediction heads. However, we emphasize that these components are deliberately designed to be minimal, and the core contribution of UniversalPose lies in the extensive parameter sharing across modalities. As described in Section 3.1 and Table 6 in appendix, more than 90\% of the total parameters are shared across all modalities through a single unified encoder.
>
> Importantly, Table 4 demonstrates that by leveraging these shared parameters and performing fine-tuning together with a small number of modality-specific parameters trained from scratch, UniversalPose can improve performance across different sensing modalities. This indicates that knowledge transferred through the shared weights effectively contributes to improving performance across modalities. Therefore, although a new modality may require training a lightweight tokenizer and head, the majority of the model remains reusable, which significantly differs from conventional pipelines that require training a separate model per modality.
>
> **(W.3) On the Value of Using Multiple Modalities.**
> We agree with the reviewer that adding more modalities does not automatically guarantee better performance, and not all modalities necessarily contribute equally to accuracy. However, in the joint-training setting, UniversalPose offers two key advantages: (i) it enables learning from diverse sensing modalities in a single training run, and (ii) it does so with a compact model, since all modalities share a single encoder rather than requiring separate models. As discussed in the related work section (“Comparison of Modalities in Human Pose Estimation.”), different modalities have different strengths and practical use cases. Having a single architecture that can support these heterogeneous modalities at once is therefore beneficial in real-world scenarios where sensing setups vary or evolve.
>
> Regarding multi-modal fusion, prior work such as X-Fi [1] has already demonstrated the effectiveness of leveraging multiple modalities. Consistent with these findings, our experiments (Table 5) show that combining four modalities with UniversalPose yields clear performance gains compared to using individual modalities alone.
> [1] X.Chen et al, "X-FI: A MODALITY-INVARIANT FOUNDATION MODEL FOR MULTIMODAL HUMAN SENSING", in ICLR2025

---

> > ### Author Response · Authors · 2025-11-23
> >
> > **(W.2) (Q.1) On single-modal performance and the effect of joint multi-modal training.**
> > To clarify the role of the unified architecture versus the amount of training data, we report in Table 1 below the results of UniversalPose instances trained separately for each modality (“Ours (Separate)”) using all subjects, alongside the jointly trained model (“Ours (Joint)”) for comparison. This comparison shows that joint training does not hurt single-modal accuracy: for depth we even observe a clear improvement, and for the other modalities the performance of “Ours (Joint)” is on par with “Ours (Separate)”, while yielding a single universal model that can handle all modalities.
> >
> > Furthermore, by comparing “Ours (Separate)” against the expert baselines in the main paper (Table 2), we see that UniversalPose already matches or surpasses many modality-specific architectures even when trained on a single modality.
> > This indicates that the gains also reflect the effectiveness of the UniversalPose architecture itself.
> > While there remain a few cases where highly specialized models slightly outperform UniversalPose, the overall trend is that UniversalPose provides competitive single-modal performance and additionally benefits from joint multi-modal training.
> >
> > ### Table 1
> >
> > | Model | Style    | WiFi (i) $\downarrow$ | mmWave $\downarrow$ | LiDAR $\downarrow$ | Depth $\downarrow$ | TSP $\downarrow$ | WiFi (ii) $\downarrow$ | Avg $\downarrow$ |
> > |-------|----------|-----------------------|----------------------|--------------------|---------------------|------------------|------------------------|------------------|
> > | Ours  | Separate | 0.191                 | 0.126                | 0.113              | 0.126               | 0.094            | 0.100                  | 0.125            |
> > | Ours  | Joint    | 0.191                 | 0.127                | 0.112              | 0.111               | 0.097            | 0.102                  | 0.123            |
> >
> > Comparison of separate vs. joint training for UniversalPose across sensing modalities.
> >
> > **(Q.2) On “scratch” vs “retrained” and single-modal performance.**
> > In Table 4 and Figure 4, both the “scratch” and “retrained” settings are trained using only the target downstream dataset for the unseen modality (e.g., depth or BGM). The only difference is the initialization: “scratch” starts from random weights, whereas “retrained” initializes the UniversalPose model from the jointly pretrained UniversalEncoder.
> >
> > Because the UniversalEncoder uses a modality-agnostic architecture, its weights learned from other modalities can be reused when adapting to a new modality. We believe that this helps capture abstract priors about human pose (e.g., kinematic structure and typical motion patterns), as well as how pose changes appear in different signal domains – for example, the relationship between pose dynamics and frequency-domain representations (joint training on TSP and WiFi, downstream on BGM), or between motion and depth structure (joint training on mmWave and LiDAR, downstream on depth). These cross-modal priors explain why fine-tuning (“retrained”) improves performance over training from scratch on the same data.
> >
> > We acknowledge that, on the single-modality benchmarks, UniversalPose still underperforms highly specialized models such as A2J and BGM2Pose. We attribute this gap to the strong modality-specific inductive biases embedded in these expert architectures: A2J employs an anchor proposal branch and in-plane offset and depth estimation branches specifically tailored for depth, while BGM2Pose uses a frequency-wise attention module carefully designed for acoustic signals. These designs rely on substantial manual engineering and domain knowledge for each modality. In contrast, UniversalPose intentionally minimizes modality-specific components in order to provide a single unified architecture that can handle diverse sensing modalities. This design reduces the engineering burden for each new sensing setup and makes the framework more scalable when extending to additional modalities. We will revise the paper to clarify this design trade-off and to make the corresponding limitations more explicit.

---

> > > ### Author Response · Authors · 2025-11-27
> > > **Additional results on more diverse modality combinations.**
> > >
> > > Motivated by feedback raised in another review, we extended our multi-modal fusion experiments with UniversalPose to evaluate performance under **a wider range of modality combinations**.  Following X-Fi, we train on the full set {Depth (D), LiDAR (L), Radar (R), WiFi (W)} with random modality dropout and then evaluate on arbitrary combinations of the four modalities using the MM-Fi dataset. In the main paper, we report results using a fixed combination of all four modalities for both training and inference, whereas this experiment specifically investigates generalization performance over more diverse modality combinations. Table 2 below compares UniversalPose and X-Fi across many modality combinations.
> > >
> > > **Table 2: Modality-fusion Comparison of UniversalPose and X-Fi (Unit: m)**
> > >
> > > | Modalities | MPJPE (Ours) | MPJPE (X-Fi) | PA-MPJPE (Ours) | PA-MPJPE (X-Fi) |
> > > | :--- | :---: | :---: | :---: | :---: |
> > > | D | **0.0841** | 0.1018 | 0.0578 | **0.0484** |
> > > | L | **0.0702** | 0.1671 | **0.0466** | 0.1032 |
> > > | R | **0.121** | 0.1274 | **0.0611** | 0.0698 |
> > > | W | **0.196** | 0.2256 | **0.105** | 0.1053 |
> > > | D+L | **0.0704** | 0.1025 | **0.0484** | 0.0484 |
> > > | D+R | **0.0766** | 0.0980 | 0.0523 | **0.0473** |
> > > | L+R | **0.0692** | 0.1098 | **0.0459** | 0.0634 |
> > > | D+W | **0.0844** | 0.1018 | 0.0581 | **0.0481** |
> > > | L+W | **0.0699** | 0.1595 | **0.0466** | 0.1027 |
> > > | R+W | **0.113** | 0.1172 | **0.0620** | 0.0627 |
> > > | D+L+R | **0.0695** | 0.0960 | 0.0476 | **0.0473** |
> > > | D+L+W | **0.0701** | 0.1020 | 0.0482 | **0.0481** |
> > > | D+R+W | **0.0767** | 0.0970 | 0.0523 | **0.0471** |
> > > | L+R+W | **0.0693** | 0.1074 | **0.0459** | 0.0631 |
> > > | D+L+R+W | **0.0693** | 0.0976 | 0.0475 | **0.0474** |
> > >
> > > We observe that UniversalPose outperforms X-Fi on the majority of combinations and metrics. These results indicate that (i) the local self-attention mechanism with pseudo 3D coordinate in UniversalPose is effective for multi-modal fusion, and (ii) updating a shared encoder during fusion training, rather than optimizing only a fusion module on top of frozen encoders as in X-Fi, can lead to higher overall accuracy.
> > >
> > > Taken together, the following capabilities establish UniversalPose as a highly general and practical framework for active-sensing–based human pose estimation:
> > >
> > > * **Handling multiple modalities** via asynchronous joint training (Table 3 in main paper);
> > > * **Improving accuracy on unseen modalities** through cross-modal fine-tuning (Table 4 in main paper); and
> > > * **Performing effective multi-modal fusion** across diverse modality combinations (Table 2 above).

---

### Official Review · Reviewer_bGJ8 · 2025-11-01

**Soundness:** 3
**Presentation:** 3
**Contribution:** 3
**Rating:** 4
**Confidence:** 4

**Summary:**

UniversalPose presents a unified Transformer framework for 3D human pose estimation that processes five heterogeneous sensing modalities (LiDAR, mmWave, WiFi, depth, acoustics) through a single shared-parameter encoder. The method converts all inputs into token sequences using lightweight modality-specific tokenizers and processes them with a universal encoder employing pseudo-3D positional embeddings and locality-aware self-attention.

**Strengths:**

+ The paper is well written and easy to follow.

+ The paper provides extensive benchmarking across 5 modalities and 8 datasets. The proposed unified framework achieves competitive or state-of-the-art performance on several single modalities (e.g., mmWave, LiDAR) and demonstrates a powerful ability for cross-modal transfer and few-shot adaptation to new modalities.

+ The unified representation enables state-of-the-art multi-modal fusion through simple token concatenation, outperforming more complex fusion baselines like X-Fi. This elegantly shows the power of the learned shared representation space.

**Weaknesses:**

- While the application to active sensing is novel, the overarching concept of a unified Transformer for multi-modal data has been explored in other domains (e.g., Omnivore). The paper could more clearly delineate its conceptual advance beyond this application shift.

- There is the complete lack of computational efficiency analysis. The paper claims "scalability" and "efficiency" as key benefits but provides no data on inference speed (FPS), FLOPs, latency, or a comparison of computational cost against deploying multiple specialized experts. Given the O(n²) complexity of self-attention with concatenated tokens, this omission severely undermines the assessment of the practical utility and real-world deployment trade-offs of the proposed method.

- The paper showcases successes but does not systematically analyze failure cases or limitations. There is no discussion of scenarios where the unified model underperforms modality-specific experts, how performance degrades with corrupted or missing modalities, or the specific conditions that lead to the training instability mentioned in Section 3.3. This leaves the reader without a clear understanding of the method's boundaries.

- The comparison is thorough against modality-specific experts and the X-Fi fusion model, but it lacks a comparison to other unified backbone architectures (e.g., an adapted Omnivore) to prove that the specific technical choices (pseudo-3D embeddings, local attention) are the key drivers of performance.

Overall, the paper presents an ambitious and technically sound unification framework with good results across a wide range of sensing modalities. However, more analysis such as computational analysis and robustness analysis are needed. I would upgrade the rating if all the concerns could be addressed and there is no novelty concerns raised by other reviewers.

**Questions:**

- In which specific scenarios or conditions does UniversalPose fail or perform worse than a modality-specific expert model? Can you provide a qualitative or quantitative analysis of these failure cases?

- The pseudo-3D coordinates are central to the method. Was the choice of a 3-dimensional space ablated? What is the performance and stability impact of using 2D or higher-dimensional spaces?

---

> ### Author Response · Authors · 2025-11-23
>
> We thank the reviewer for acknowledging the strong performance demonstrated through extensive experiments. Below we address the weaknesses (W) and the questions (Q) raised by the reviewer.
>
> **(W.1) The accuracy comparison with other unified backbone architectures.**
> The reviewer pointed out the need for a comparison with other unified architectures such as OMNIVORE [1]. We consider the “ViT-like” model variant in Table 2 to correspond to this comparison. In the original Omnivore paper, the authors state: “We use the same model (parameters) to process all the resulting embeddings. While OMNIVORE can use any vision transformer architecture to process the patch embeddings, we use the Swin transformer architecture as our base model given its strong performance on image and video tasks,” which suggests that the choice of backbone architecture is flexible. In our task, however, it is not obvious how to use Swin Transformer in a unified manner for point clouds (mmWave, LiDAR), audio, and WiFi CSI. Therefore, we regard a ViT-like model with global self-attention applied to each modality as an adapted Omnivore-style baseline for comparison. As shown in Table 2, our proposed method substantially outperforms this baseline, indicating that combining pseudo 3D coordinates with local attention can achieve high accuracy across diverse modalities.
>
> [1] R. Girdhar et al., “OMNIVORE: A Single Model for Many Visual Modalities”, CVPR 2022.
>
> **(W.2) Computational efficiency comparison.**
> Table 1 below compares the MACs of our UniversalPose with the modality-specific models measured by the THOP library. As shown in the table, UniversalPose has MACs on the same order as the heaviest expert models, while replacing five separate encoders with a single shared backbone. We believe this is because, although UniversalPose uses self-attention, it restricts attention to k-nearest neighbors and thus performs local rather than fully global attention, keeping the MACs comparable to the heaviest modality-specific baseline.
> We therefore view the computational cost as reasonable given the gain in flexibility and unification across modalities.
>
> _Table 1: MACs (in G) of modality-specific baselines and UniversalPose for each sensing modality._
>
> | Model                     | mmWave (G) | LiDAR (G) | WiFi-CSI (G) | Depth (G) | TSP (G) |
> |---------------------------|-----------:|----------:|-------------:|----------:|--------:|
> | Modality-specific baselines |     0.350 |     18.4  |        15.9  |     10.1  |   0.849 |
> | UniversalPose             |      14.3  |     19.1  |        14.3  |     14.3  |   14.3  |
>
> **(W.3)(Q.1) Limitations and boundary analysis.**
> We thank the reviewer for highlighting the importance of a clearer discussion of failure cases and limitations. We acknowledge that UniversalPose does not always outperform modality-specific expert models in terms of raw accuracy. As shown in Table 3 and Table 4 of the main paper, our model underperforms A2J on depth-based pose estimation and BGM2Pose on acoustic-based pose estimation.
>
> We attribute this performance gap to the strong modality-specific inductive biases embedded in these expert models. For instance, A2J incorporates specialized components such as an anchor proposal branch and in-plane offset and depth estimation branches, carefully designed for depth signals. Similarly, BGM2Pose employs a frequency-wise attention module tailored to the structure of acoustic signals. These architectures leverage extensive manual engineering and domain knowledge that directly optimize for a single modality.
>
> In contrast, UniversalPose intentionally minimizes modality-specific design choices. Our goal is to provide a single unified architecture that can handle diverse sensing modalities, rather than to hand-tune architectures for each modality. To the best of our knowledge, UniversalPose is the first framework that achieves competitive performance across such heterogeneous inputs while avoiding heavy manual engineering for each modality. We will revise the paper to make these design trade-offs and the corresponding failure cases more explicit.

---

> > ### Author Response · Authors · 2025-11-23
> >
> > **(Q.2) On the dimensionality of the pseudo-3D coordinate space.**
> > We appreciate the reviewer’s insightful question regarding the choice of a three-dimensional pseudo-coordinate space. The dimensionality was not selected due to any strict theoretical constraint, but rather as a pragmatic design choice motivated by the inherent nature of certain modalities, particularly point clouds, which are natively represented in 3D physical space.
> >
> > Moreover, increasing the dimensionality beyond 3 would require additional transformations to map raw sensor outputs (e.g., mmWave, LiDAR) into a higher-dimensional positional embedding space, introducing extra modality-specific projection layers that run counter to our goal of keeping manual modality engineering minimal. To avoid this additional complexity, we fix the pseudo-coordinate space to 3D for all experiments in the paper.

---

> > > ### Author Response · Authors · 2025-11-27
> > > **Additional results on more diverse modality combinations.**
> > >
> > > Motivated by feedback raised in another review, we extended our multi-modal fusion experiments with UniversalPose to evaluate performance under **a wider range of modality combinations**.  Following X-Fi, we train on the full set {Depth (D), LiDAR (L), Radar (R), WiFi (W)} with random modality dropout and then evaluate on arbitrary combinations of the four modalities using the MM-Fi dataset. In the main paper, we report results using a fixed combination of all four modalities for both training and inference, whereas this experiment specifically investigates generalization performance over more diverse modality combinations. Table 2 below compares UniversalPose and X-Fi across many modality combinations.
> > >
> > > **Table 2: Modality-fusion Comparison of UniversalPose and X-Fi (Unit: m)**
> > >
> > > | Modalities | MPJPE (Ours) | MPJPE (X-Fi) | PA-MPJPE (Ours) | PA-MPJPE (X-Fi) |
> > > | :--- | :---: | :---: | :---: | :---: |
> > > | D | **0.0841** | 0.1018 | 0.0578 | **0.0484** |
> > > | L | **0.0702** | 0.1671 | **0.0466** | 0.1032 |
> > > | R | **0.121** | 0.1274 | **0.0611** | 0.0698 |
> > > | W | **0.196** | 0.2256 | **0.105** | 0.1053 |
> > > | D+L | **0.0704** | 0.1025 | **0.0484** | 0.0484 |
> > > | D+R | **0.0766** | 0.0980 | 0.0523 | **0.0473** |
> > > | L+R | **0.0692** | 0.1098 | **0.0459** | 0.0634 |
> > > | D+W | **0.0844** | 0.1018 | 0.0581 | **0.0481** |
> > > | L+W | **0.0699** | 0.1595 | **0.0466** | 0.1027 |
> > > | R+W | **0.113** | 0.1172 | **0.0620** | 0.0627 |
> > > | D+L+R | **0.0695** | 0.0960 | 0.0476 | **0.0473** |
> > > | D+L+W | **0.0701** | 0.1020 | 0.0482 | **0.0481** |
> > > | D+R+W | **0.0767** | 0.0970 | 0.0523 | **0.0471** |
> > > | L+R+W | **0.0693** | 0.1074 | **0.0459** | 0.0631 |
> > > | D+L+R+W | **0.0693** | 0.0976 | 0.0475 | **0.0474** |
> > >
> > > We observe that UniversalPose outperforms X-Fi on the majority of combinations and metrics. These results indicate that (i) the local self-attention mechanism with pseudo 3D coordinate in UniversalPose is effective for multi-modal fusion, and (ii) updating a shared encoder during fusion training, rather than optimizing only a fusion module on top of frozen encoders as in X-Fi, can lead to higher overall accuracy.
> > >
> > > Taken together, the following capabilities establish UniversalPose as a highly general and practical framework for active-sensing–based human pose estimation:
> > >
> > > * **Handling multiple modalities** via asynchronous joint training (Table 3 in main paper);
> > > * **Improving accuracy on unseen modalities** through cross-modal fine-tuning (Table 4 in main paper); and
> > > * **Performing effective multi-modal fusion** across diverse modality combinations (Table 2 above).

---

### Meta-Review · Area_Chair_3xT5 · 2025-12-24

**Summary:**

UniversalPose is a unified transformer framework that estimates 3D human pose from five sensor types including WiFi, LiDAR, and acoustics. It converts diverse signals into tokens and processes them using a shared encoder with pseudo 3D positional embeddings.

Strengths:

1. The model supports joint training on unpaired data and performs well in multi-modal fusion.

2. The paper is well written and includes extensive benchmarking across multiple datasets.

Weaknesses:

1.Several reviewers noted that while the encoder is shared, the architecture still requires MLP tokenizers for each sensor.

2. A unified transformer for multi-modal data has been explored previously in other domains (e.g., Omnivore), and the current application shift may not represent a substantial conceptual advance.

3. Using many modalities is not always necessary if fewer sensors can achieve similar accuracy.

**Reviewer Concerns:**

A major concern is whether the performance gains come from the architecture itself or simply from training on a larger volume of data. Additionally, the concept of a unified transformer for multi-modal data is seen as having limited novelty since it has been explored in other domains.

**Reviewer Scores:**

The ratings are 6, 4, 4, and 2, which suggests a lean toward rejection. I believe that reviewers remain unconvinced about the true innovation and the necessity of the modality specific components.

---

### Decision · Program_Chairs · 2026-01-26

Reject